# Stakeholders’ Consensus on Strategies for Self- and Other-Regulation of Video Game Play: A Mixed Methods Study

**DOI:** 10.3390/ijerph17113846

**Published:** 2020-05-28

**Authors:** Michelle Colder Carras, Matthew Carras, Alain B. Labrique

**Affiliations:** 1Department of International Health, Johns Hopkins Bloomberg School of Public Health, 615 N. Wolfe St. W5501, Baltimore, MD 21205, USA; 2Carras Consulting, 1407 Scanlan Drive, Glen Burnie, MD 21205, USA; matthew.carras@gmail.com; 3Departments of International Health, Epidemiology, and Health Policy and Management, Johns Hopkins Bloomberg School of Public Health, 615 N. Wolfe St. W5501, Baltimore, MD 21205, USA; alabriq1@jhu.edu

**Keywords:** gaming disorder, prevention, video games, mixed methods research, stakeholder engagement, consensus development

## Abstract

*Background:* Little is known about strategies or mechanics to improve self-regulation of video game play that could be developed into novel interventions. This study used a participatory approach with the gaming community to uncover insider knowledge about techniques to promote healthy play and prevent gaming disorder. *Methods:* We used a pragmatic approach to conduct a convergent-design mixed-methods study with participants attending a science fiction and education convention. Six participants answered questions about gaming engagement and self- or game-based regulation of gaming which were then categorized into pre-determined (a priori) themes by the presenters during the presentation. The categorized themes and examples from participant responses were presented back to participants for review and discussion. Seven participants ranked their top choices of themes for each question. The rankings were analyzed using a nonparametric approach to show consensus around specific themes. *Results:* Participants suggested several novel potential targets for preventive interventions including specific types of social (e.g., play with others in a group) or self-regulation processes (e.g., set timers or alarms). Suggestions for game mechanics that could help included clear break points and short missions, but loot boxes were not mentioned. *Conclusions:* Our consensus development approach produced many specific suggestions that could be implemented by game developers or tested as public health interventions, such as encouraging breaks through game mechanics, alarms or other limit setting; encouraging group gaming; and discussing and supporting setting appropriate time or activity goals around gaming (e.g., three quests, one hour). As some suggestions here have not been addressed previously as potential interventions, this suggests the importance of including gamers as stakeholders in research on the prevention of gaming disorder and the promotion of healthy gaming. A large-scale, online approach using these methods with multiple stakeholder groups could make effective use of players’ in-depth knowledge and help speed discovery and translation of possible preventive interventions into practice and policy.

## 1. Introduction

Research on problems with video game play has led to a proposed diagnosis of Internet gaming disorder (IGD) in the Diagnostic and Statistical Manual 5 [1] and inclusion of gaming disorder in ICD-11 [2]. Representative surveys usually show that only a small percentage of the general population has clinical problems related to excessive gaming [3]. Systematic reviews link gaming disorder to correlates such as impulsivity, male gender, impaired mental health and low psychosocial well-being [3,4,5]. However, the Goldilocks hypothesis (based on the fairytale character Goldilocks, who struggles to find sizes of chair, porridge and bed that are “just right) suggests that engaging with technology in a moderate way is not harmful [6]. The WHO notes that most people are able to play video games without problems [7]. As this suggests that moderate video game play is a reasonable public health goal, investigating how people can play in a healthy way is vital to support the public health objective of preventing gaming disorder.

While many studies examine motivations to play, less is known about how gamers self-regulate their gaming. One qualitative study in children uncovered several reasons for engaging (e.g., motives, habit, and contextual factors) or disengaging (e.g., self-regulation, parental intervention, or disruption/frustration of gaming outcomes or psychological needs; [8]). Contextual factors of games or the environment also contribute to the length of gaming sessions: gaming sessions may be prolonged through losing track of time [9], achievement, winning or game mechanics, including gambling-like mechanisms such as loot boxes [8,10]. Disengaging from play may also be easier with support from real-life friends [11] or when using strategies like setting an alarm, taking regular breaks, or setting goals in the game [9]. The above findings show a range of individual, game and contextual factors that can promote healthy gaming and prevent disorder.

However, policy responses have focused largely on limiting time gaming [12] or regulating loot boxes as gambling [13]. Commentaries suggest that existing tools for self-regulation of gambling, such as self-exclusion, mandatory breaks, or pop-up messaging, might be useful for gaming [14]. A few technology platforms have already implemented feedback mechanisms to alert players about time or allow them to limit their time [15,16]. Games also provide such mechanisms in the form of parental controls. Some also remind players to take breaks, but forcing shutdown or exclusion periods as a preventive mechanism has so far been limited [12]. The current focus on a few limited policy aspects also neglects the need to take into account cultural factors, individual player factors, and how games change over time [12].

With the recent inclusion of gaming disorder in ICD-11, calls for collaboration with industry stakeholders have increased. Many researchers have suggested that the game industry has a responsibility to protect the health of gamers [14,17,18]. Fewer researchers support including gamers as partners in research and policy efforts to understand or prevent gaming disorder [19,20].

Participatory and multi-stakeholder approaches in health research allow for better translation of research into practice [21,22]. Stakeholders are members of diverse groups, including experts and the general public, who can provide unique perspectives on an issue. In the case of gaming disorder, this could mean not just clinicians, researchers, and policy makers, but also gamers, their families, and people who create games. Stakeholder-engaged research that involves group deliberation and decision-making promotes the type of co-learning that allows researchers to better examine all aspects of a problem or question, encourages decision-makers to consider multiple perspectives, and makes science more transparent [23,24]. Using systematic methods that integrate perspectives of those in the know—in this case video game players and developers—with research and theory will be vital to allow research on problematic gaming to address insider perspectives and cultural differences and keep up with changing technologies.

This approach has been used with gamers to explore their unique insights into what it means to be “addicted” [19]. As an illustration of the usefulness of this approach, we discovered that a small sample of gamers at a gaming fan convention were not only interested in discussing game “addiction”, their ideas overlapped only partially with criteria for IGD. In fact, their suggestions leaned toward less strict criteria for social impairment and they suggested appointment mechanics (a game feature promoting engagement) as a primary contributor to loss of control. These findings suggest that including gamers in discussions of research into gaming disorder may improve research. However, a systematic, perspective-integrating approach that examines factors associated with gaming self-regulation would better contribute to intervention development.

To address this gap and provide clinicians, decision-makers, the game industry and the general public with additional insights about gaming self-regulation, we conducted participatory, mixed-methods, convergent-design research with gamers to discuss the challenges of regulating video game play. This research builds on our prior study regarding gamers’ insights on game “addiction” [19]. The qualitative objectives of our study were to elicit a wide variety of knowledge from the video gaming stakeholder community about why gamers play, why they stop playing, and how they self-regulate gaming time and to map these to known themes in the literature. The quantitative goal was to numerically rank the consensus on importance of themes. The overall mixed methods goal was to demonstrate a rapid and transparent consensus-development approach that can be used with multiple stakeholders. Such an approach would foster knowledge development and research co-creation in order to develop and prioritize research and prevention interventions for problematic video game play.

## 2. Materials and Methods

### 2.1. Overview

Mixed methods research combines qualitative and quantitative data to answer research questions that require a deep understanding of multiple perspectives, contexts, and/or communities/cultures [25]. We used a convergent mixed methods design, i.e., we combined qualitative and quantitative data simultaneously, to develop a rapid consensus on the relative importance of themes related to video gaming engagement and self-regulation. We used a pragmatic approach toward study design and analysis, as these allow researchers to combine diverse approaches to best answer the research question and place equal value on both objective and subjective knowledge [26]. In our previous study [19], for example, we used an emic, free-listing approach, working with our gamer audience at a panel to generate categories and themes for free-listed responses. As this proved difficult within the one-hour time frame of a panel at a convention, we opted to combine deductive (pre-determined themes from the literature) and inductive (free-listing) approaches in the qualitative part of our study design here.

As qualitative research benefits from researchers’ self-reflection, we describe our relevant background here. All authors are researchers with experience in conducting mixed methods research in technology and health, particularly in video games. We have conducted a similar study in another setting to address gamers’ insights into game “addiction” [19]. The first two authors, who conducted the panel, also identify as members of the gaming community, regularly attend gaming fan conventions, and have personal experience with regulating their gaming time. They use this personal experience to connect with stakeholders and facilitate meaningful discussion of healthy and unhealthy video game play. Author MCC is a researcher specializing in video game play, technology use and mental health; author MC is an information technology specialist with game industry experience; and author AL is a professor specializing in digital health.

To address our mixed method goal of demonstrating a rapid and transparent consensus-development technique, we first used a deductive approach among authors to identify prominent themes around self-regulation and gaming from the literature (Section 2.2). We then used an emic, inductive, free-listing approach to gather data about self- and other-regulation of video game play from audience members at a presentation (Section 2.4). We then facilitated a group decision-making procedure (Delphi process) with the audience to brainstorm, discuss/debate, and finally quantitatively rank concepts related to gaming self-regulation and steps the gaming industry could take to facilitate healthy play. We present the qualitative, quantitative and mixed methods results to better understand self-regulation as gamers see it. Our presentation here was guided by American Psychological Association standards for mixed methods and qualitative research reporting [27,28].

### 2.2. Selection of Themes for Framework

Prior to the presentation, the authors first briefly reviewed relevant literature on game engagement [8,9,29] and design [29,30,31,32]. We extracted themes and frameworks from these into tables and bulleted notes and discussed and combined results to condense themes surrounding game engagement and self-regulation. This deductive approach was not meant to be exhaustive but to present the most salient themes for discussion; other themes could be incorporated during data collection if necessary. This condensed list of themes (Table 1) was used to facilitate rapid thematic analysis during presentation and discussion with audience stakeholders. Themes were transcribed to index cards placed on the table in front of the presenters and used in a deductive, top-down theoretical thematic analysis approach [33] to rapidly classify participant responses during the presentations.

### 2.3. Participants

A panel was presented at a science fiction and education convention, Escape Velocity, held in Washington, D.C. in 2017. The convention featured topics and events relevant to science and technology, STEAM education, video games, and pop culture. The conference is sponsored by the Museum of Science Fiction and science organizations like NASA, nonprofit organizations and industry groups, and in its first year was attended by over 2000 people [34].

Participants were recruited passively through panel descriptions in the convention schedule and through posters on the door to the panel room. Our panel on Day 1 was entitled “Insights Needed: Video Gaming and the Goldilocks Principle” and our panel on Day 2 was entitled “Insights Needed Wrap-Up: Give Us Your Feedback on Video Games and Engagement Preliminary Analysis.” The authors had no prior relationship with the research participants. Each session began with the reading of a consent document where participants were offered the opportunity to leave if they did not want to participate; no audience members left. Participants did not receive compensation for participation. As the pragmatic study design was dependent on the setting of a panel at a convention, we did not identify a needed number of participants ahead of time and our sample was one of convenience. One participant arrived late and two participants did not complete all portions of the questionnaire.

Six respondents completed at least some demographic information and percentages are reported out of those who answered the question. Age ranged from 27 to 49 (Mean = 35, SD 7.9) and one participant was female. Most respondents (66.7%) were white, one identified as mixed race, and one declined to answer. Half had a graduate degree and another 33.3% had graduated from college. All respondents identified as members of the gamer stakeholder group, but also identified with other groups such as family/friend of a gamer (n = 3), researcher (n = 3), educator (n = 1), and industry (n = 1). In response to the question “Do you see yourself as a gamer”, all responded either Yes or “Kind of (intermediate/casual)” but one participant also answered “No” to the question “Do you play video games, including mobile games (e.g., Candy Crush, Pokémon GO)?”. (This participant identified primarily as a member of the game industry.) All worked full time and the majority (80%) earned over $40,000/year. The group played on average 3.13 h a day (SD 1.31) and 5.33 days per week (SD 1.15). Half played about 1–2 h per session while the other half played 2–3 h per session.

### 2.4. Data Collection and Study Procedure

Questionnaire responses, sticky notes used in free listing, and field notes made up the qualitative and quantitative data as described here. This approach described is a combination of nominal group technique and Delphi approach that the authors used in a previous study [19]. Prior to data collection the authors printed out questionnaires with questions about demographics, stakeholder group membership, gaming experience and ranking of themes and added a number to each questionnaire. We then created a set of sticky notes with matching participant numbers. Each set of sticky notes contained five sticky notes for each of our four questions. Each note was labeled with a participant number and Q1, Q2, Q3, or Q4.

Data were collected over the course of an hour at a panel discussion. At the beginning of the panel, the presenters (the first two authors) introduced themselves as researchers and gamers and described the presentation as a study using group decision-making techniques to help answer questions about how to achieve just the right amount of engagement with video games. We then asked participants to list ideas for promoting healthy gaming by asking four open-ended questions (Table 2) about gaming engagement and regulation strategies. Participants recorded answers on sticky notes, then completed paper questionnaires asking about demographics and gaming experience. We allowed 10 min for data collection. Participants were asked not to complete the questions about ranking the importance of themes until told to do so.

As data on sticky notes was collected, responses were categorized by the presenters by placing each sticky note under a pre-determined theme’s index card. Disagreements about how to categorize responses were resolved through discussion between presenters. Once all responses were categorized (about 10 min after all responses were received) they were transcribed into the slideshow and displayed to all participants. We asked for clarifications of some responses and, as a participant check, discussed whether we had categorized responses for each question appropriately based on our own understanding as gamers and researchers. Field notes were incorporated into the response examples at times for further clarification. Participants elaborated on some responses, disagreed with one another or provided additional explanation, and a robust discussion of specific responses and themes ensued. The authors’ goal was to ensure clarity of themes and their examples; this was accomplished with about three minutes of discussion for each question.

Participants were then asked to rank the top three themes they felt best answered each question by listing their choices for most important themes in order on the last page of the questionnaire, e.g., Question 1: #1 = immersion, #2 = game type, #3 = achievement. The numeric ranking participants gave to themes for a particular question constituted the quantitative data. Participants were invited to return in two days for the results. After the panel, one author transcribed questions, responses, and themes into an Excel spreadsheet to record qualitative data and the results of thematic analysis. Another author entered rankings and questionnaire responses directly into a Stata do-file to create and analyze a quantitative analysis database. To validate data quality, data entry was double-checked and analyses re-run prior to manuscript revision.

At a second panel two days later, we discussed the study’s organizing frameworks and procedures and presented the results of the ranking. We again asked for clarifications of some responses and further discussed difficult-to-categorize responses. Finally, we presented the limitations and strengths we felt the process provided and invited participants to provide additional feedback. Data are available as Appendix A.

### 2.5. Qualitative and Quantitative Analysis

#### 2.5.1. Categorization of Responses

We categorized responses as the unit of analysis during the presentation using a deductive approach based on the a priori themes. We chose theory-based thematic analysis as a practical way to facilitate rapid categorization in keeping with our pragmatic approach. Initial thematic analysis was performed at the panel and final categorizations and themes for each question were arrived at after discussion with participants; we used participant checks to facilitate categorization of some responses. Themes and examples (Table 2) were then projected onscreen to allow participants to rank them.

#### 2.5.2. Calculation of Ranks

We used an observational design to gather participants’ perspectives on the importance of themes for each question. Ranked themes were assigned a numerical value from one to eleven to represent the eleven total themes (the nine a priori themes from Table 1 and two themes emerging during group discussion). We validated data entry against collected data by listing database results for each question in a table and checking against each participant’s questionnaire.

For each question, we transformed a participant’s ranked themes into coded rankings by replacing a placeholder variable for each theme with the numeral 1, 2, or 3 if a participant ranked it among their top three choices. This resulted in a unique subset of themes as the domain for each question. If a respondent did not rank a particular theme for that subset, the rank for that theme for that participant was coded as 0. We conducted a quantitative consensus analysis of participants’ ranking of the most important themes for each question by calculating the weighted sum of centered ranks (WSCR) in Stata 13 IC [35] using the Skillings–Mack statistic (skilmack.ado, [36]). This flexible approach aggregates rankings across participants by combining the ranking for a specific question provided by each participant with the consensus on ranking for that question across all participants. With this approach, high consensus themes (those that are ranked more often or ranked very highly) will have a high and positive WSCR while low consensus themes, i.e., those that are ranked fewer times, ranked as less important, or show greater disagreement among respondents will have a low or negative WSCR. As we used rankings rather than raw numerical values in the analysis and replaced non-ranked themes with 0, standard errors are the same for all resulting WSCR for each question.

### 2.6. Presentation of Results

Qualitative results from the rapid thematic analysis are presented first, then qualitative and quantitative results are presented together to reflect the convergent design. Table 3 combines ranked themes with examples and important topics of discussion to illustrate how group processes affected participants’ choices. We used these results to suggest ways that multiple stakeholder groups could help promote healthy gaming (Figure 1). This report follows standards for reporting qualitative, quantitative, and mixed methods research established by the American Psychological Association [27,28].

### 2.7. Methodological Integrity

To ensure fidelity and utility of our qualitative findings and address our research objectives, we address the adequacy, grounding, and context of data and how we used it to develop meaningful and coherent findings. First, we collected data from cultural insiders—those who had an interest in video gaming and self-regulation—grounding our data in their perspective. We used an emic approach to data collection, free listing, to elicit their cultural understanding and insider knowledge. We further grounded findings in evidence by using free-listed responses to illustrate themes. We included all audience members in our analysis, since all members identified as gamers. However, audience members also identified as members of other stakeholder groups. Although our sample was small, we aimed to gather insights/knowledge from gamers and other stakeholders; we feel the collected data are therefore adequate to capture a broad range of perspectives. Researchers’ perspectives were used to determine an initial set of themes, facilitate rapid data analysis and guide group discussion.

We demonstrate consistency in our analysis by describing consensus development and discussion and when and how they were used in the analytic process. We used data displays (tables), author consensus on a priori themes, group discussion with participants, formal quantitative consensus, and participant feedback as checks to promote rigor in the qualitative analysis. We discuss the context of data collection and present findings coherently, reflecting on discrepancies and reporting according to mixed methods standards. Our results further the understanding of how gamers and other stakeholders can make insightful and meaningful contributions to a clinical- and policy-relevant research.

### 2.8. Ethics

The study protocol was approved by the institutional review board of the Johns Hopkins Bloomberg School of Public Health (IRB No. 00006931) and carried out in accordance with the Declaration of Helsinki. The need for written informed consent was waived by the institutional review board. Participants were fully informed about the study at the beginning of each study session and offered the opportunity not to participate.

## 3. Results

Six people participated in the free listing round and submitted between one and three answers per question. Seven people participated in the ranking round and six people completed demographic and gameplay questionnaires. Four questionnaires were complete, including rankings, while two were partially complete.

### 3.1. Qualitative

We initially identified nine themes from our brief review of several articles related to gaming engagement and design [8,9,29,30,31,32]. Our selection of themes was not meant to be exhaustive of all themes in the gaming literature, but rather a starting point for our discussion-based study design. Themes were chosen based on the authors’ knowledge of game engagement and game design and how these factors contribute to problematic or self-regulated gaming. Categorization of responses to our four questions resulted in three to six themes for each question (Table 2). Some responses transcended themes, such as pop-up windows being a behavioral game mechanic that was “game-breaking”, thus primarily having to do with the theme of immersion (Q4). Through discussion, some submissions categorized by the authors as having to do with one theme were reassigned to a different theme and one novel theme, analyzing moves, was suggested but did not end up being ranked highly by participants. Another theme, competition, was suggested as being different from achievement. The pre-determined theme of habit was not seen to match any of the responses. The full database of responses and assigned themes are available as Appendix A.

### 3.2. Mixed Methods Synthesis

Table 3 presents results of nonparametric quantitative analysis that describe consensus on themes as they were ranked by participants. Immersion (e.g., *exploration of dynamic universe*) and achievement (e.g., *rewards*) emerged as important features of games that motivated playing (Q1), but there was less agreement on game type, mood and competition. Social factors (e.g., *online interactions with terrible people*) and novelty (e.g., *repetition/grinding with little variety*) were considered equally important reasons to stop playing (Q2). Immersion (e.g., *games that are crashy or slow*) was also agreed to be somewhat important, but achievement (*games that are not balanced*), environment (*out of time*), and mood (*not fun anymore*) were considered less important.

Social factors showed up again in Q3 as a strategy people could use to self-regulate (e.g., *play with others in a game that requires everyone to be present*). One response categorized as social described how online group play could limit gaming time. Participants debated this initially, but after deliberation agreed that choosing to play games online in groups might be an effective way to reduce time as other group members’ leave, disrupting the group. Self-regulation strategies were also considered important. Responses focused on setting firm goals (*just three quests*) or limiting time (*setting an alarm, set a schedule for gaming time that is after your other work/responsibilities have been finished*) or money spent on gaming.

For Q4, there was strong agreement that the game industry could make it easier for players to self-regulate by changing or adding behavioral game mechanics (e.g., *clear break points, rewards for short sessions, negative reinforcement like XP decay*). Participants also offered social (*taking away features of online multiplayer and have local play only*) or environment-related (e.g., *integrated health tips into game that use health points that players can use themselves*) strategies, but these were less often ranked. The group also suggested benefits to new technology such as augmented reality games (e.g., Pokémon GO) that would offer novelty (*local multiplayer fun with new tech*), but there was low consensus on this theme. The group discussed this with regard to the benefits of staying grounded in the real world, especially when playing with others, which was thought to be useful for reducing the feeling of immersion that could lead to time loss.

## 4. Discussion

This study used a systematic, consensus-developing participatory approach with members of the gaming community to explore the importance gamers assigned to themes about self-regulating gaming. We found that even a small group (n = 7) was able to offer many specific suggestions that could form the basis of preventive interventions (Figure 1). These results support the emphasis on game mechanics and self-regulation identified in previous research [8,9,19], but add an additional dimension, social factors, that presents a new potential focus of prevention research. The findings also contribute to the importance of incorporating cultural insider knowledge about gaming and self-regulation into research and intervention development across disciplines.

The fact that gamers endorse the theme of self-regulation so strongly suggests that integrating specific techniques for promoting self-regulation (Figure 1) into preventive interventions may be a worthwhile approach. These strategies may be more palatable when endorsed by the gaming community itself. One possibility would be to have celebrity gamers (e.g., popular Twitch streamers) suggest or even discuss such strategies with their gaming audience. Hearing about risk from trusted individuals is important, and recent research shows that the general public may place greater trust in social media personalities than experts who may offer conflicting or confusing messages [37]. Streamers do address mental health topics in Twitch [38], so this may become an important mode for delivering preventive interventions relating to video game play.

Our small sample had many specific suggestions about features to include or leave out of games, such as exit points—places where they could leave the game without feeling that more had to be done. Exit points have been specifically mentioned as a factor of ethical game design that can be promoted in the video game industry [39]. Exit points may be unique to games and were not included in a recent expert-developed list of recommended tools for the prevention of problematic gambling that could be revised to address gaming disorder [14].

Another specific suggestion was to incentivize taking breaks. Our sample mentioned how games sometimes punish players for limiting their play, for example temporarily blocking players from team play if the leave a team abruptly. While games may do this to foster engagement, this also makes it more difficult for gamers to leave when competing opportunities or demands arise, such as a friend’s request for a visit or the dawning recognition that time has passed and the dishes still need to be washed. Games can also reward breaks by manipulating experience gains, either by allowing players to progress faster when they return from a break or limiting progress after long sessions. While interventions related to experience gains have been implemented in several countries (experience decay), this approach may be less appealing to game developers [12]. In contrast, allowing faster progress when returning from a break (e.g., rested experience) may be a more palatable way to reward healthy amounts of gaming.

Social factors are an area brought up by our participants that seems to have little mention in suggestions for prevention interventions. Our sample found them important as a driver of disengaging (negative online interactions) and self-regulating (playing with others in a game that requires everyone to be present). In fact, this theme is a good example of the strength of a group reflection process as it reflected a single participant’s suggestion. At the beginning of the discussion, most participants didn’t understand or agree with the suggestion that choosing to “play with others in a game that requires everyone to be present…since the other players can effectively force you to stop” would be a good way to self-regulate, but after a few minutes of discussion, six out of seven people ranked that as one of the most effective ways to limit time. This reveals the unique benefit to conducting a Delphi method approach with gamers as stakeholders—the perspective of an individual can be clarified within a group and ultimately recognized as a ground truth: Yes, if we choose to play games with that require a group, our time may be limited whether we choose to limit it (or are able to effectively limit it) or not.

Our findings suggest that focusing research and prevention efforts without involving insights from community members may unnecessarily limit the range of potential interventions decision-makers consider and impede progress toward implementation. The current epistemological approach toward research on gaming disorder places greater emphasis on clinical expertise rather than multiple perspectives [40]. The findings here challenge this traditional approach by considering gamers as vital co-creators of knowledge. For example, loot boxes have been the focus of much public health activity related to gaming disorder to date [13,41], yet loot boxes or other gambling-like mechanics were not mentioned by our sample. Other targets may be more palatable to multiple stakeholders, potentially easier to achieve, or more effective at preventing problematic gaming. Gamers are also useful in pointing out nuances of how we think about interventions. For example, while research may concentrate on immersion as a motivating factor for problematic gaming, our applied research here allowed gamers themselves to make a clear link between an example intervention (pop-ups) and the experience of losing immersion (pop-ups as game-breaking).

While this study was limited in sample size, it does reflect the gender (male) and age of gamers in the United States [42,43]. The small size does limit generalizability, and other sample makeups may suggest different responses. For example, our sample was self-selected for attending a science fiction and education convention and participating in a panel focused on gaming research. Other samples may prioritize different themes when ranking questions related to self-regulation. Our sample also felt the theme of Immersion to be more important than achievement—had we used a sample of gamers at a gaming tournament, for example, this would likely be different. In addition, our sample was English-speaking and based in the US, so findings may not transfer to other countries or cultures. However, one goal of this study was to demonstrate that the combination of perspective seeking, deliberation and systematic quantitative consensus analysis provides additional information beyond that of a survey, interview or focus group. Further research would benefit from systematic recruitment of a large sample of various stakeholders and a formal, multi-round consensus development technique in an online format to encourage transparency.

## 5. Conclusions

Formal consensus development techniques with gamers produced many specific examples of ways gamers can self-regulate and also emphasized the importance of game mechanics in fostering healthy gaming and preventing gaming disorder. These examples are supported by previous research and could serve as the basis for interventions to promote healthy gaming (Figure 1). Going forward, decision makers could increase the value of research on the prevention of problematic or disordered gaming by involving gamers, developers, clinicians and other stakeholders in co-learning about self-regulation strategies, game mechanics, and other features of game engagement.

## Figures and Tables

**Figure 1 ijerph-17-03846-f001:**
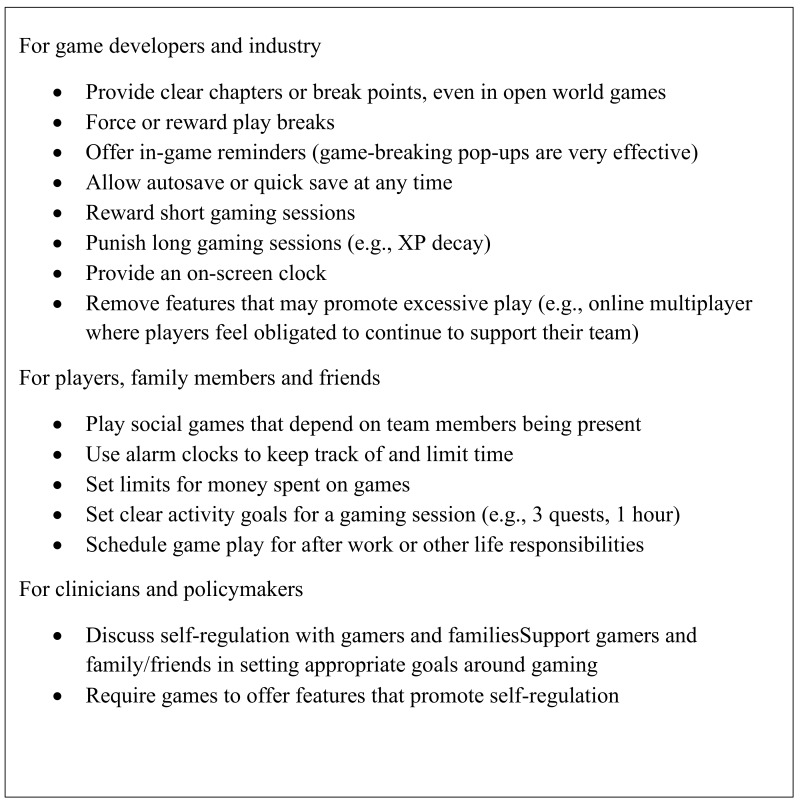
Suggestions for regulating video game play.

**Table 1 ijerph-17-03846-t001:** Prepared themes and examples used in thematic analysis.

Theme	Example Theme/Concept from Literature [8,9,29,30,31,32]
Achievement	Non-competitive. Also includes general success in completing objectives
	Achievement, challenge, Advancement, Mechanics, Being stronger, Rare items, Meta-game rewards
Behavioral game mechanics	Game-driven regulation, Alerts, Behavioral tracking, Exit points, Timed rewards, Fast loading times
Environmental	Forces outside the scope of the game’s design
	Content-driven, family/parent driven, outside needs & responsibilities
Habit	Habit
Immersion	The feeling of being one with the game
	Narrative, Look and feel, Interface, Autonomy, Customization, Role-playing, Discovery, Fantasy
Mood regulation	Arousal/enjoyment, seeking diversion or distraction, escapism
Novelty	New or interesting aspects of the game
Self-regulation	Self-regulation
Social	Social interaction, relationships, teamwork

**Table 2 ijerph-17-03846-t002:** Themes and examples presented for ranking.

Theme	Example 1	Example 2	Example 3
Q1 What features of games make you want to play them or keep playing?			
Competition	*Winning*		
Game type	*Arcade games*		
Immersion	*Known intellectual property that you like*	*Customizing*	
Mood regulation	*Repeatable enjoyment*		
Achievement	*Well-balanced gameplay*	*Rewards*	
Q2 What features of games make you want to stop playing?			
Social	Playing online with terrible people	*Losing*	
Environmental	*Out of time*		
Immersion	*Broken games*	*Games that crash*	*Unclear control of avatar*
Achievement	*Games that are not balanced*	*Hard learning curve*	
Mood regulation	*Not fun anymore*		
Novelty	*Repetition/grinding*		
Q3 What are some strategies people can use to regulate the amount of time they game?			
Self-regulation	*Set an alarm*	*Set goals for gameplay*	
Social	*Play with others to limit time*		
Analyze moves	*Slower-move analytically*		
Q4 What are some things the game industry could do to make it easier to regulate the amount that people play games?			
Game mechanics	*Game-breaking pop-ups*	*Provide break points*	*On screen clock*
Environmental	*Integrate health tips*		
Social	*Local multiplayer only*		
Novelty	*Local multiplayer—AR fun with new tech*		

**Table 3 ijerph-17-03846-t003:** Ranked themes with examples of responses (n = 7)**.**

Theme as Categorized by Authors	Times Ranked ^a^	WSCR ^b^	WSCR/SE	Response Examples
Q1. What features of games make you want to play them or keep playing?
Immersion	5	7.78	1.47	*Visual/audio aesthetics, exploration of dynamic universe, known theme or IP, player/playstyle customization, great stories, color, sound*
Achievement	5	3.54	0.67	*Rewards, either literal ones in the form of items, unlocks, in-game money, OR psychological rewards such as a rewarding story pay off or resolution to a puzzle; well-balanced gameplay*
Game type	4	0	0	*Arcade-based games like racers or mech games.*
Mood	3	−0.71	−0.13	*Repeatable enjoyment*
Competition ^c^	1	−10.61	−2.00	*Winning*

Q2. What features of games make you want to stop playing?
Social	4	5.24	0.89	*Online interaction with terrible people (most)*
Novelty	5	5.24	0.89	*Repetition/grinding with little variety*
Immersion	4	1.96	0.33	*Unclear control of avatar, Games that are crashy or slow*
Environment	2	−3.93	−0.66	*Out of time*
Achievement	2	−3.93	−0.66	*Unfair or unreliable game mechanics that result in “cheap” deaths/failures, games that are not balanced, games that are broken/not working*
Mood	3	−4.58	−0.77	*Not fun anymore*

Q3. What are some strategies people can use to regulate the amount of time they game?
Self-regulation	7	5.2	1.39	2x^d,c^-*Limit time…* (*60 min/session*),2x^d^-*Use an alarm clock*, 2x-*Set firm goals for gameplay ex. “Just 3 quests”, set a schedule for gaming after other responsibilities*
Social	6	0	0	*Play with others in a game that requires everyone to be present…since the other players can effectively force you to stop*
Analyzing moves ^c^	5	−5.2	−1.39	*Analyzing moves, moving slower*

Q4. What are some things the game industry could do to make it easier to regulate the amount that people play games?
Mechanics	6	10.07	2.2	*Clear delineations between missions/chapters which offer clear “break points” to the player, as opposed to open-ended/open world games which don’t give an “excuse” to stop, autosave/quicksave, rewards that emphasize short sessions or have negative reinforcement mechanisms like XP delay, time allowed*
Environment	5	−0.77	−0.17	*Integrated health tips into game that use health points that players can use themselves*
Social	5	−2.32	−0.51	*Local multiplayer only- take away features of MO*
Novelty	3	−6.97	−1.52	*Augmented Reality Gaming—local multiplayer fun with new tech*

^a^ Number of times an item was ranked as one of a participant’s top three choices. ^b^ Weighted sum of centered ranks statistic from the Skillings–Mack statistic, where high/positive ranks correspond to agreed-upon items and low/negative ranks correspond to items that were chosen less often or ranked as less important. SE were the same for each question: SE_Q1_ = 5.29, SE_Q2_ = 5.92, SE_Q3_ = 3.74, SE_Q4_ = 4.58. ^c^ New theme emerging from discussion. ^d^ More than one version/wording submitted by different respondents.

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
