# Peer review of "Stakeholders’ Consensus on Strategies for Self- and Other-Regulation of Video Game Play: A Mixed Methods Study"

_ijerph, 2020, doi:10.3390/ijerph17113846_

Round 1

Reviewer 1 Report

This is an excellent and well written contribution to the literature. I only have three minor suggestions: (1) report number of participants in the abstract already, (2) mention the specific suggestions mentioned in the conclusions section in the abstract, and (3) move descriptive statistics from 3. to 2.3.

Author Response

This is an excellent and well written contribution to the literature. I only have three minor suggestions: (1) report number of participants in the abstract already, (2) mention the specific suggestions mentioned in the conclusions section in the abstract, and (3) move descriptive statistics from 3. to 2.3.

Thank you for these suggestions. We have made the recommended changes in (1) the abstract, (2) the results and conclusions sections of the abstract and (3) section 2.3, pg 5 line 151.

Reviewer 2 Report

The paper adopts a mixed method to gather subjective information how to promote healthy play and prevent gaming disorder. The introduction of the research is well-written. The second part of the study needs a strong revision.
The authors should describe the objectives of study.
2.1. Overview. This section should be revised. I think it describes the analysis of the study.
2.2. Selection of themes for framework. The content of this section is also hard to understand.
2.3. Participants. Please provide more information about participants such as age, gender, etc.
2.4. Procedure. This section should include more information where data were collected.
2.5. Qualitative and quantitative analysis. Qualitative analysis requires several steps that authors did not mentioned in their plan of statistical analysis. In addition, no information was provided about the quantitative analysis.
The paper needs a strong revision. Author should follow the APA guidelines to write the paper.

Author Response

The paper adopts a mixed method to gather subjective information how to promote healthy play and prevent gaming disorder. The introduction of the research is well-written. The second part of the study needs a strong revision.

Thank you for your comments. We appreciate your attention to detail and offer specific responses below.

The authors should describe the objectives of study.

Thank you for this suggestion. We draw the reviewer's attention to the text on lines 100-107, where we describe the qualitative, quantitative and mixed-methods objectives  of the study.

2.1. Overview. This section should be revised. I think it describes the analysis of the study.

This section (2.1) was revised and expanded according to JARS-Qualitative standards.

2.2. Selection of themes for framework. The content of this section is also hard to understand.

Thank you for this feedback. We provided additional detail on the process for selecting themes in Section 2.2. Hopefully this will clarify the process.

2.3. Participants. Please provide more information about participants such as age, gender, etc.

This information was previously found in section 3, but on another reviewer's suggestion we have moved it up to section 2.3. We have also expanded our description according to JARS-Qual in that same section.

2.4. Procedure. This section should include more information where data were collected.

We have expanded section 2.4 to include much more detail on data collection according to JARS-Qualitative, and have renamed the section to Data Collection and Study Procedure.

2.5. Qualitative and quantitative analysis. Qualitative analysis requires several steps that authors did not mentioned in their plan of statistical analysis. In addition, no information was provided about the quantitative analysis.

Thank you for drawing attention to this—we agree that more thorough reporting is useful. Our original intention was to focus on meeting the guidelines for mixed methods reporting which we mentioned originally in section 2.6. We had also geared our reporting toward prevention implications, which reflects our pragmatic approach. We are happy to expand on the methods and have done so with guidance from the APA reporting standards (JARS-Qual and JARS-Quant) iin addition to the previously-used JARS-Mixed Methods. The new reporting has led to changes in each subsection of the Methods section and the addition of another subsection, 2.7, Methodological Integrity. We hope this satisfies these suggestions.

The paper needs a strong revision. Author should follow the APA guidelines to write the paper.

We thank the reviewer for this suggestion, which we have taken careful note of. As our audience for this paper is primarily public health practitioners, decision-makers, researchers and clinicians, we strived to balance reporting standards with brevity and clarity. In this revision we have meticulously attended to each element of reporting standards from the APA JARS website. This has led to extensive revisions throughout the Methods section, the Qualitative (Section 3.1)  and Mixed Methods (3.2) sections of the results, and additional inclusions in the Discussion section on p 14 line 33-34, p 15 line 53 and lines 81-84, and page 17 lines 94-100.

Reviewer 3 Report

This study used a participatory approach with the gaming community to uncover insider knowledge about techniques to

This is an interesting study from the viewpoints of video game players. It provides practicable strategies for promote healthy play and prevent problematic gaming.

Some minor suggestions are proposed for improving the manuscript.

  1. The authors should introduce the definition of “stakeholder” in this study. There are many roles in video gaming community. The term “stakeholder” lacked clearly defined.
  2. Please add more introductions for the methods that the authors used to promote discussion among participants and researchers in the present study.
  3. I agree that collaboration with industry stakeholders is necessary to develop strategies of preventing problematic video gaming. Please provide some suggestions based on the results of the present study for the policy makers and industry stakeholders.

Author Response

This is an interesting study from the viewpoints of video game players. It provides practicable strategies for promote healthy play and prevent problematic gaming.

Thank you for your comments. We appreciate your attention to our manuscript and offer specific responses below.

Some minor suggestions are proposed for improving the manuscript.

  1. The authors should introduce the definition of “stakeholder” in this study. There are many roles in video gaming community. The term “stakeholder” lacked clearly defined.

We added a definition of stakeholder on lines 78-81.

  1. Please add more introductions for the methods that the authors used to promote discussion among participants and researchers in the present study.

We clarified this further in section 2.4, Data collection and study procedure and now report according to APA standards (JARS Mixed Methods, Qualitative and Quantitative).

  1. I agree that collaboration with industry stakeholders is necessary to develop strategies of preventing problematic video gaming. Please provide some suggestions based on the results of the present study for the policy makers and industry stakeholders.

Thank you for your support. We draw the reviewer's attention to Figure 1, which contains bulleted lists for game developers and industry; players, family members and friends; and clinicians and policymakers. We now also include some specific suggestions in the Abstract.

Round 2

Reviewer 2 Report

The current version of the manuscript is much improved. The authors should be congratulated for this.

Author Response

Thank you for your comments-we appreciate your suggestions!